# (-)-Epigallocatechin-3-Gallate Prevents IL-1β-Induced uPAR Expression and Invasiveness via the Suppression of NF-κB and AP-1 in Human Bladder Cancer Cells

**DOI:** 10.3390/ijms232214008

**Published:** 2022-11-13

**Authors:** Dhiraj Kumar Sah, Pham Ngoc Khoi, Shinan Li, Archana Arjunan, Jae-Uk Jeong, Young Do Jung

**Affiliations:** 1Department of Biochemistry, Chonnam National University Medical School, Hwasun 58128, Republic of Korea; 2Faculty of Basic Medical Sciences, Pham Ngoc Thach University of Medicine, Ho Chi Minh City 740500, Vietnam; 3Department of Radiation Oncology, Chonnam National University Medical School, Hwasun 58128, Republic of Korea

**Keywords:** AP-1, cell invasion, EGCG, IL-1β, Urokinase-type plasminogen activator receptor (uPAR), NF-κB, ROS

## Abstract

(-)-Epigallocatechin-3-O-gallate (EGCG), a primary green tea polyphenol, has powerful iron scavengers, belongs to the family of flavonoids with antioxidant properties, and can be used to prevent cancer. Urokinase-type plasminogen activator receptors (uPARs) are glycosylphosphatidylinositol (GPI)-anchored cell membrane receptors that have crucial roles in cell invasion and metastasis of several cancers including bladder cancer. The mechanism of action of EGCG on uPAR expression has not been reported clearly yet. In this study, we investigated the effect of EGCG on interleukin (IL)-1β-induced cell invasion and uPAR activity in T24 human bladder cancer cells. Interestingly, nuclear factor (NF)-κB and activator protein (AP)-1 transcription factors were critically required for IL-1β-induced high uPAR expression, and EGCG suppressed the transcriptional activity of both the ERK1/2 and JNK signaling pathways with the AP-1 subunit c-Jun. EGCG blocked the IL-1β-stimulated reactive oxygen species (ROS) production, in turn suppressing NF-κB signaling and anti-invasion effects by inhibiting uPAR expression. These results suggest that EGCG may exert at least part of its anticancer effect by controlling uPAR expression through the suppression of ERK1/2, JNK, AP-1, and NF-κB.

## 1. Introduction

Cancer is characterized by uncontrolled cell growth, abnormal differentiation, proliferation, invasion, and metastasis [1]. The bladder is a hollow organ in the lower pelvis with flexible, muscular walls that can stretch to hold urine and contract to send it out of the body. The development of bladder cancer is caused by uncontrolled growth of cells in the urinary bladder. The diagnosis of bladder cancer is divided into noninvasive bladder cancer (NIBC) and invasive bladder cancer (IBC). NIBC is responsible for approximately 70% of bladder cancer cases, whereas IBC is more malignant and has a five-year survival rate of less than 35% [2]. Bladder cancer affects men three times more commonly than women, with a 1-in-26 lifetime risk of developing the disease among men and 1-in-88 among women. Caucasians have a two-fold increased risk of bladder cancer compared with black people or Hispanic people, but this type of cancer tends to be diagnosed at an advanced stage, making it harder to cure [3].

In the last three decades, several research studies have explored the molecular mechanisms of cancer development. Even though cancer is multifactorial, numerous in vivo and in vitro studies have demonstrated that chronic inflammation has a significant impact on the development of this disease and patient survival [4]. Inflammatory markers such as leukocytes, cytokines, chemokines, cyclooxygenases, prostaglandins, reactive oxygen, and nitrogen species (ROS/RNS), and metalloproteinases cause genetic/epigenetic alterations in normal cells. These modifications harm the cells’ DNA, prevent it from being repaired, alter transcription factors, inhibit apoptosis, and promote angiogenesis, which results in the development of cancer [5]. These inflammatory mediators may thus have a role in developing into cancer biomarkers at all stages of the disease.

The key cytokine in both humoral and cellular inflammation is interleukin-1 (IL-1). IL-1, which is secreted by cancer cells, stromal components, or infiltrating leukocytes, and plays a role in the modulating anti-tumor immunity during the progression of cancer [6]. One of the key members of the IL family, IL-1β, has pleiotropic effects on immune cells, angiogenesis, cancer cell growth, migration, and metastasis [7,8]. In addition to the inflammatory response, IL-1β can also induce urokinase-type plasminogen activator (uPA) and matrix metalloproteinase (MMP) expression, thus, promoting proteolysis of the extracellular matrix in different cells through the PKCα, MAPK, and NF-κB pathways [9]. The uPARs are GPI-anchored cell membrane receptors involved in the degradation of the extracellular matrix in colorectal cancer and are with an early stage and very poor prognosis [10]. Recently, uPARs have been reported to be involved in metastasis, angiogenesis, inflammatory cell chemotaxis, and degradation of the extracellular matrix [11].

Worldwide, cancer is the leading cause of death and one of the biggest burdens on societies due to a lack of resources for prevention and effective treatment. In patients with non-muscle-invasive disease and localized muscle-invasive disease, radical cystectomy (RC) with lymphadenectomy is the treatment of choice [12]. A staggering 50–70% of patients die from their cancer within five years of undergoing surgery. Predictably, this is due to microscopic metastatic disease prior to cystectomy [13]. Thus, biomarkers are needed for patients with bladder cancer to select the correct treatment and follow-up these patients individually [14].

Researchers have extensively investigated the anticarcinogenic effects of green tea. The polyphenol content of green tea extract is composed of (-)-epigallocatechin gallate (EGCG), (-)-epigallocatechin (EGC), (-)-epicatechin gallate (EGG), and (-)-epicatechin gallate (ECG) [15]. Furthermore, green tea catechins are also powerful iron scavengers that belong to the family of flavonoids with antioxidant properties that can be used to prevent cancer [16]. EGCG is significantly protective against bladder degeneration in rats and acts as an antioxidant against H_2_O_2_-induced oxidative stress through superoxide production in normal and malignant bladder cells [17]. In a study, EGCG reduced the levels of phospho (p)-epidermal growth factor receptor (EGFR), H-RAS, p-RAF, p-MEK1/2, and p-ERK1/2 in human thyroid carcinoma cells [18]. EGCG does not appear to have anti-invasion effects on bladder cancer cells, and its mechanism of action has not been reported clearly yet. We examined the molecular mechanisms by which EGCG inhibits IL-1β-induced uPAR expression in our current study. A new medicinal strategy for muscle invasive bladder cancer (MIBC) treatment could be developed by demonstrating that EGCG suppresses uPAR expression by blocking the MAPK pathway (ERK1/2, JNK)-induced transcriptional activity of AP-1 and NF-κB. This could be the first step toward developing a drug that inhibits MIBC invasion after treatment.

## 2. Results

### 2.1. uPAR Expression and Correlation in Bladder Cancer

An analysis of the overall survival of patients with bladder cancer was conducted using the GEPIA2 database to verify the accuracy of uPAR and IL1B expressions (Figure 1A). A 50% cutoff was set for high and low values. Statistical significance was defined as *p* < 0.05. The gene expression of PLAUR and IL-1β is shown in bladder carcinoma. The expression of IL-1β in bladder cancer is 2.71 transmission per million (TPM) and 2.4 TPM in normal samples. uPAR expression in bladder cancer is 4.99 TPM and 4.08 TPM in normal samples (Figure 1A). Figure 1B shows the logarithmic uPAR expression in bladder carcinoma. These results indicated that there was a significantly increased uPAR expression in non-papillary urothelial carcinoma. Figure 1C shows the uPAR expression in various stages of BLCA. Figure 1D shows the significant expression correlation of IL-1β, NF-κB1, JUN, and MAPK1 with PLAUR (TIMER2.0). The GeneMANIA results are as follows for the network data of the abovementioned genes: physical interaction: 77.64%, co-expression: 8.01%, predicted: 5.37%, co-localization: 3.63%, pathways: 1.88%, and shared protein domains: 0.60% (Figure 1E).

### 2.2. EGCG Suppresses uPAR Upregulation Induced by IL-1β

To determine how EGCG inhibits the upregulation of uPAR by IL-1β, RT-PCR and Western blotting was used to evaluate the mRNA and protein level after IL-1β was treated with T24 cells at various concentrations for 4 h and 12 h, respectively. We showed that uPAR expression was dose-dependently increased by IL-1β (Figure 2A), as described in a previous study [9]. After 5 ng/mL IL-1β was added to T24 cells that had previously been pretreated with EGCG at concentrations of 5–50 μM, uPAR transcriptional and protein levels were detected using RT-PCR and Western blotting (Figure 2B). A promoter-luciferase assay was used to further study the inhibitory effect of EGCG on uPAR expression. The outcomes (Figure 2C) additionally demonstrated that EGCG reduced the IL-1β-activated uPAR promoter in a concentration-dependent manner. The present study’s use of EGCG (0–50 μM) did not affect the viability of T24 cells (Appendix A).

### 2.3. EGCG Inhibits IL-1β-Induced uPAR Expression via the Inhibition of MAPK Signaling

In our earlier research, we showed that the primary signal mediates the IL-1β-induced overexpression of uPAR in the T24 cell line for various periods with ERK1/2 and JNK [9]. Using chemical inhibitors SB (P38 inhibitor), PD (ERK1/2 inhibitor), and SP (JNK inhibitor) that specifically block uPAR-activated P38, ERK1/2, and JNK signaling, the results of an RT-PCR experiment revealed that ERK1/2 and JNK activation largely activates AP-1, a crucial transcription factors for controlling uPAR expression (Figure 3A). As a result, we speculated that EGCG might have an impact on the MAPK signaling that IL-1β activates. As evidenced by the findings in Figure 3B, EGCG suppressed ERK1/2 and JNK activation during the 1 h pretreatment followed by 30 min of IL-1β administration. To investigate the underlying mechanism further, we exposed human T24 cells to IL-1β in the presence or absence of the MAPK inhibitor. T24 cells treated with JNK inhibitor SP and ERK1/2 inhibitor PD decreased IL-1β-induced c-Jun phosphorylation (Figure 3C), and the results of an AP-1 promoter assay using chemical inhibitors (PD, and SP) (Appendix A) indicate that ERK1/2 and JNK may interact upstream of AP-1 during the IL-1β-augmented uPAR expression. This finding suggests that EGCG suppresses IL-1β-activated MAPK (ERK1/2 and JNK) pathways that are crucial to uPAR induction.

### 2.4. EGCG Suppresses IL-1β-Induced AP-1 Activation

As shown above, AP-1 transcription aspects are critical for the IL-1β induction of uPAR expression. We then investigated their contributions to the suppression of uPAR expression caused by EGCG. RT-PCR analysis showed that EGCG inhibited IL-1β-activated AP-1 dose-dependently (Figure 4A). Since c-Jun is one of the components of AP-1, we also examined the effect of EGCG on c-Jun by Western blotting to obtain further insight into the mechanism of the EGCG-mediated downregulation of AP-1 (Figure 4B). We found that EGCG inhibited the IL-1β-induced increased phosphorylation of c-Jun. To strengthen the results, we determined the effect of EGCG by AP-1-dependent transcription studies and showed that IL-1β-activated AP-1 was dose-dependently suppressed by EGCG (Figure 4C). These results indicated that EGCG reduces IL-1β-induced uPAR via the suppression of AP-1 activation.

### 2.5. EGCG Inhibits NF-κB Activity That is Involved in IL-1β-Induced uPAR Upregulation

NF-κB is constitutively active during cancer initiation, development, and metastasis, presumably as a result of the inflammatory microenvironment and oncogenic mutations [19]. Numerous studies have shown that NF-κB and AP-1 are the most important factors in regulating uPAR expression in human cancer cell lines and tumors [9,20,21]. NF-κB activation is typically accompanied by an increase in IκBα phosphorylation; as expected, IL-1β enhanced the activation of phosphorylated NF-κB and phosphorylated IκBα, which was abrogated by EGCG pretreatment (Figure 5A). Consequently, we investigated the role of NF-κB in EGCG-induced inhibition and IL-1β-induced uPAR overexpression in T24 cells. In addition, an NF-κB promoter assay determined that EGCG suppressed the NF-κB promoter activity stimulated by IL-1β in a dose-dependent manner (Figure 5B). The results in Figure 5C show that Bay11-7082, a selective inhibitor of NF-κB signaling, substantially inhibited IL-1β-induced uPAR overexpression at the mRNA and protein (Appendix A) level. NAC, which is an antioxidant drug that increases glutathione levels and scavenges free radicals, suppresses NF-κB activation induced by several extracellular stimuli, including oxidative stress [22,23]. After testing the role of NAC by Western blotting to detect NF-κB expression, NAC was found to decrease the phosphorylation of NF-κB and its subunit IκBα in T24 bladder cancer cells (Figure 5D). These results indicated that EGCG inhibits NF-κB downstream signaling pathways, similarly to NAC.

### 2.6. Involvement of ROS in EGCG-Induced Inhibition IL-1β-Induced uPAR Expression

To determine whether EGCG inhibits ROS production induced by IL-1β in human bladder cancer T24 cells, the level of ROS was measured using DCFDA, a fluorophore that is sensitive to H_2_O_2_. We observed that EGCG pretreatment effectively inhibited the ROS produced by IL-1β, which are reported to be substantially stimulated by IL-1β (Figure 6A,B). ROS are well known for stimulating the transcription factor NF-κB [24], as shown in Figure 5D, which is the inhibitory effect of NAC. Before IL-1β treatment, T24 cells were pretreated with NAC to examine the specific roles of ROS in IL-1β-induced uPAR expression. NAC inhibited the IL-1β-induced mRNA expression of uPAR in a dose-dependent manner, as demonstrated in Figure 6C, and protein levels were shown in Appendix A. These findings imply that ROS production was inhibited by EGCG, preventing NF-κB activation and the additional uPAR expression induced by IL-1β.

### 2.7. EGCG Inhibits the IL-1β Stimulation of Human Bladder Cancer T24 Cell Invasiveness

The advancement of cancer is also aided by invasion, which is a significant step. We measured the impact of uPAR inhibitors on the quantity of migrating cells invading through a modified Boyden invasion chamber to assess the impact of EGCG and IL-1β on tumor invasion. The quantity of invading cells that pierced through the Matrigel barrier increased after T24 cells were incubated with IL-1β. We discovered that pretreating the cells with EGCG or an anti-uPAR antibody reduced the number of cells that pierced the Matrigel barrier (Figure 7A). In addition, before being exposed to IL-1β, T24 cells were pre-treated with anti-uPAR antibodies, non-specific IgG, and various concentrations of EGCG. After being incubated with uPAR-neutralizing antibodies and EGCG, the IL-1β-treated cells largely lost their Matrigel invasiveness; however, exposure to non-specific IgG and IL-1β had no discernible effect on the level of cell invasiveness (Figure 7B). Based on these findings, it is likely that through the suppression of IL-1β-induced uPAR expression, EGCG partially prevented a cell invasion.

## 3. Discussion

Globally, cancer accounted for almost 10.0 millions of deaths in 2020, rendering it the leading cause of death [25]. Chemotherapy is the most common cancer treatment used in conventional medicine to treat malignant cancer; in contrast, localized lesions are dealt with by surgery and radiotherapy [26]. Several natural products may be used in cancer chemoprevention and treatment to inhibit tumor growth [27,28]. Green tea is one of the world’s most popular beverages due to its powerful antioxidants. Green tea has been thoroughly studied for its anticancer and chemotherapeutic properties [29]. Green tea use has been linked to a moderate decrease in the risk of malignancies such as prostate [30], colorectal [31], and esophageal cancer in epidemiological studies [29]. A major component of green tea’s antioxidants is polyphenols, including phenolic acids and catechins.

Catechins, which are found in food and medicinal plants, have significant anticarcinogenic activities that have drawn the attention of many researchers [32]. EGCG is the major constituent of catechins present in green tea and has received considerable attention in scientific research as a potential agent for cancer treatment and prevention. However, research continues to be conducted into the therapeutic effects of EGCG in cancer prevention and treatment. In the current work, we describe the mechanism of action of EGCG in T24 bladder cancer cells under the presence of IL-1β, which is demonstrated to inhibit tumor proliferation and uPAR upregulation. This study will improve the potential of EGCG to target human bladder cancer cells.

An inflammatory cytokine, IL-1β, influences immune cells, proliferation, migration, angiogenesis, and metastasis in cancer [33,34]. In this study, GEPIA2 and TIMER2.0 results showed significantly increased IL-1β and uPAR expression in bladder cancer subjects. Ryuji et al. reported that increased IL-1β expression in Aldo-keto reductase 1C1 (AKR1C1) induced bladder cancer cell lines [35]. uPAR expression is critical in tumorigenesis, and high endogenous uPAR levels have been linked to tumor proliferation and advanced metastatic tumors; glycolytic capacity has also been linked to the angiogenesis of various solid and hematologic malignancies [36]. In addition, several therapeutic strategies have been developed to inhibit the functions of uPAR in clinical trials and preclinical studies. Some synthetic inhibitors may serve as promising therapeutic targets by inhibiting uPAR; however, no apparent effect has been demonstrated [37]. Previous studies have shown that high inducers of uPAR expression in cancer cells such as prostaglandin E2, cadmium, nicotine, lithocholic acid, and lysophosphatidic acid activate uPAR by various molecular pathways [38,39,40,41,42], and Lee et al. demonstrated that IL-1β stimulates the expression of uPAR in gastric cancer [43].

Herein, we demonstrated that EGCG could inhibit this stimulating action of IL-1β on bladder cancer cells. Siddiqui et al. found that EGCG inhibits the expression of vascular endothelial growth factor (VEGF), uPA, and angiopoietins 1 and 2 in prostate cancer cell lines [44]. The molecular docking results showed that EGCG decreased uPA activity by binding to urokinase and inhibiting His 57 and Ser 195 of the urokinase catalytic triad while extending to Arg 35 from a positively charged urokinase loop [45]. EGCG decreased p-AXL, ALDH1A1, and SLUG levels in tumors while preventing the development of tumors in mice [46]. EGCG inhibited tumor growth in vivo by downregulating the expression of miR-25 and proteins associated with apoptosis, which was validated by decreased Ki-67 levels and increased pro-apoptotic PARP levels [47]. EGCG suppresses recepteur d’origine nantais (RON) expression in gastric cancer cells by inhibiting Egr-1 (-) [15] and blocked MMP-9 expression induced by nicotine in endothelial cells [48]. As shown in the present study, EGCG inhibits IL-1β-induced uPAR upregulation, suggesting that EGCG delays cancer progression by inhibiting the inflammatory response.

NF-κB and AP-1 are transcription factors that regulate several genes involved in inflammation, embryonic development, lymphoid differentiation, oncogenesis, and apoptosis [49]. The transcription factor NF-kB is widely expressed and is implicated in inflammatory and immunological responses such as uPAR. Our finding (Figure 5C) also indicated that BAY inhibits the uPAR expression induced by IL-1β. NF-κB dimers are kept inactive in cytosol by binding to IκB proteins, which inhibits them. IκB kinase (IKK) is activated by IL-1β, thereby phosphorylating, ubiquitinating, and degrading IκBs, facilitating the translocation of the free NF-κB dimer into the nucleus [50]. Several studies have shown that EGCG inhibits NF-kB signaling in various cancers [51,52,53]. EGCG significantly suppressed IL-1β-mediated IRAK degradation as well as signaling cascades downstream from IRAK degradation including IKK activation, IκBα degradation, and NF-kB activation via reducing the phosphorylation of the NF-kB p65 subunit [54], indicating that EGCG suppresses the expression of the NF-κB/AP-1 signaling pathway. Our findings support a similar mechanism of EGCG suppressing IL-1β-induced NF-κB and AP-1 in T24 bladder cancer cells, along with a delay in IκBα and c-Jun degradation.

MAPK is gaining popularity as a cancer preventive and treatment target molecule. The stimulation of MAPK pathways may result in the suppression of AP-1-mediated gene expression. MAPKs (ERK, JNK, and p38) can trigger several transcription factors, including ELK and c-Jun, components of AP-1, resulting in altered cell proliferation, migration, and apoptosis [45]. Figure 3 indicates that PD and SP inhibit uPAR and p-c-Jun expression but not SB. Our findings indicate that IL-1β induced MAPK (ERK, JNK) signaling in T24 cells and that EGCG blocked the signaling activation of these factors, but not p38 activity, to inhibit uPAR overexpression. Mari et al. also showed that EGCG reduced p38 MAPK activation and suppressed ERK1/2 phosphorylation in HT1080 cells [55]. The Jun and Fos dimer complex AP-1 promotes cell proliferation, differentiation, and survival by triggering cytokines, growth factors, and oncoproteins. This study demonstrated the inhibitory effect of EGCG on IL-1β-induced c-Jun expression in T24 cells. Various studies support our findings that EGCG inhibits AP-1 activation via altered AP-1 DNA binding activity, AP-1 transcriptional activity, and the inhibition of ERK1/2 activity, but not p38 kinase activity [56,57].

Similarly, ROS also have a dynamic impact on the tumor microenvironment (TME) and have been linked to cancer survival, angiogenesis, and metastasis at various concentrations. ROS initiate the cancer cell survival signaling cascade, which includes MAPK/ERK1/2, p38, JNK, and PI3K/Akt, which then activates NF-κB, MMPs, and VEGF [58]. The findings of this study indicate that EGCG inhibits H_2_O_2_-induced ROS production. Several in vitro and in vivo studies have shown that EGCG reduces ROS generation [59,60]. Furthermore, EGCG reduces ROS production by enhancing the activity of superoxide dismutase (SOD) and glutathione peroxidase (GSH-px) and by decreasing the malondialdehyde (MDA) level in H_2_O_2_-induced oxidative stress injury [61]. Additionally, NAC inhibits the ROS production by impairing NF-κB signaling (Figure 5D). NAC (N-acetyl-l-cysteine) is commonly used to identify and test ROS inducers, and to inhibit ROS. These data indicated that EGCG inhibits ROS formation by impairing the AP-1/NF-kB signaling pathways, implying that it is protective against the IL-1β-induced ROS production in bladder cancer cell lines.

On the other hand, proinflammatory IL-1β cytokines play an important role in malignant cells invasiveness. IL-1β is abundant in cancerous tissue and promotes tumor growth, invasion, carcinogenesis, and host–tumor interactions [62]. Furthermore, ROS enhance cell invasiveness by inducing uPAR expression via the ERK-1/2 and AP-1 signaling pathways [63]. These results indicate that EGCG inhibits IL-1β cell invasion. Several IL-1β mutant models have demonstrated resistance to cancer growth and reduced tumor invasion [64,65,66]. In addition, multiple signaling pathways are activated as a result of the IL-1β stimulation of tumor cells, such as the PKB, MAPK, and NF-κB pathways. These signaling molecules activate MMP-9, a matrix degrading enzyme involved in IL-1β-induced tumor invasion [67]. To support our findings of invasiveness, studies have shown that EGCG significantly inhibited the gelatinolytic activity of MMP-2 and MMP-9 [68] as well as the elastinolytic activity of MMP-12 [69]. In addition to inhibiting proMMP-2 activation, EGCG may also compete with MT1-MMP or one of the mechanisms involved in cancer invasion and metastasis [69]. EGCG was also revealed to inhibit G (1) to S-phase cell cycle progression and MMP-9 activation via the transcription factors NF-κB and AP-1 [70]. Furthermore, Cornelia et al. stated that EGCG inhibits the expression of VEGF, which may contribute to the suppression of cancer cell invasion [71]. These data clearly demonstrate the inhibitory action of EGCG on cell invasiveness induced by IL-1β in T24 cells. We designed a schematic to depict the mechanism underlying the EGCG inhibiting the IL-1β-induced uPAR expression based on all the aforementioned discoveries (Figure 8). In summary, IL-1β induces uPAR expression and activates the NF-κB/AP-1/MAPKs signaling pathways, which lead to bladder cancer progression, invasion, and metastasis.

## 4. Materials and Methods

### 4.1. Bioinformatics Analysis

We used the Gene Expression Profiling Interactive Analysis (GEPIA) Database to identify the PLAUR and IL-1β expression in BLCA from GEPIA2 (http://gepia2.cancer-pku.cn/ (accessed on 19.07.2022)) public database (404 tumors and 27 normal samples). TIMER2.0 (http://timer.cistrome.org/ (accessed on 19.07.2022)) and GeneMANIA (https://genemania.org/ (accessed on 19.07.2022)) were used to identify the correlation and network interaction between PLAUR with IL-1β, NF-κB1, JUN, and MAPK1 genes.

### 4.2. Materials and Conditions of Cell Culture

T24 human bladder carcinoma cells were obtained from the American Type Culture Collection (Rockville, MD, USA). Cultures of the cells were performed at 37 °C in 5% CO_2_ in DMEM containing 10% fetal bovine serum (FBS) and 1% penicillin-streptomycin. IL-1β was obtained from R&D systems (USA) and harvested at various intervals. EGCG was purchased from Sigma-Aldrich (St. Louis, MO, USA). Bay11-7082 (Bay), PD, SP, and SB203580 (SB) from Calbiochem (San Diego, CA, USA) and N-acetyl-L-cysteine (NAC) from Sigma were freshly prepared by dissolving in water immediately before use. The 2×10^4^ T24 cells were planted in a 96-well plate containing DMEM supplemented with various concentration of EGCG for 24 h. The cell viability was analyzed by performing an MTT assay (Sigma-Aldrich, St. Louis, MO, USA).

### 4.3. Reverse-Transcription Polymerase Chain Reaction (RT-PCR)

TRIzol (Invitrogen, Carlsbad, CA, USA) reagent was used to extract RNA from T24 cells. Invitrogen’s SuperScript reverse transcriptase was used to synthesize first-strand complementary DNA from one microgram of total RNA. The PCR master kit (iNtRON, Seongnam, Gyeonggi-do, Korea) was used to amplify the cDNA using a primer set against β-actin and uPAR genes as follows: β-actin forward, 5′-AAG CAG GAG TAT GAC GAG TC-3′ and β-actin reverse, 5′-GCC TTC ATA CAT CTC AAG TT-3′ (561 bp); uPAR forward, 5′-AAG ACC CTG AGC TAT CGG ACT G-3′; uPAR reverse, 5′- TGC ATT CGA GGT AAC GCC TCC -3′ (130 bp); C-JUN forward, 5′-GAACCCCTCCTGCTCATCTG-3′; C-JUN reverse, 5′-GGAAACGACCTTCTATCACG-3′ (316 bp).

### 4.4. Western Blot Analysis

After being washed with PBS, T24 cells were detached with trypsin, and the proteins present in the cells were extracted using whole-cell lysis buffer (PRE-PERP protein extraction kit, iNtRON, Korea). Then, 40 μg of the proteins was separated by SDS-PAGE and transferred to 0.45-μM PVDF membranes (Millipore Corporation, Billerica, MA, USA). The transferred membranes were blocked with a TBST solution containing 5% skimmed milk before being incubated with a primary antibody overnight at 4 °C. A horseradish peroxidase-labeled secondary antibody (Cell Signaling Technology, Danvers, MA, USA) was used to detect immunoreactive signals in a luminescence detection system after being rinsed three times with TBST (0.1% Tween-20 in TBS) at 10 min intervals. The primary antibodies were as follows: anti-uPAR antibody (Cell Signaling Technology, Danvers, MA, USA), antiphosphorylated NF-κB p65 antibody (Cell Signaling Technology, Danvers, MA, USA), antiphosphorylated IκBα (Ser32) antibody (Cell Signaling Technology, Danvers, MA, USA), anti-c-Jun antibody (Santa Cruz Biotechnology), antiphosphorylated c-Jun antibody (Cell Signaling Technology, Danvers, MA, USA), antiphosphorylated p38 MAPK antibody (Cell Signaling Technology, Danvers, MA, USA), antiphosphorylated ERK antibody (Cell Signaling Technology, Danvers, MA, USA), antiphosphorylated JNK antibody (Cell Signaling Technology, Danvers, MA, USA), and anti-β-actin (Cell Signaling Technology, Danvers, MA, USA) monoclonal antibodies.

### 4.5. Promoter Activity Assay

Transiently transfected T24 cells with an uPAR promoter-luciferase reporter construct (pGL3-uPAR) were acquired from Dr. Y. Wang (Australian National University, Canberra, Australia) and used to investigate the uPAR promoter activity. T24 cells were seeded and cultured until they attained a confluence of 70–80%. The plasmid containing the pGL3-uPAR promoter was then transfected into cells using FuGENE 6 (Promega, Madison, WI, USA) according to the manufacturer’s instructions. A plasmid (PRL-TK) was transfected as an internal control that contained a constitutively active Renilla luciferase reporter gene linked to the promoter of herpes simplex thymidine kinase. After one day of incubation with the transfection medium, the cells were treated with IL-1β for 4 h. Subsequently, the effect of EGCG on the activity of the uPAR promoter was assessed by pretreating the cells for 1 h with EGCG before treating them with IL-1β. Transfecting the cells with the pGL-uPAR promoter vector carrying mutant AP-1 and NF-κB binding sites allowed researchers to investigate the significance of NF-κB and AP-1 in IL-1β-induced uPAR expression. T24 cells were separated and crushed with a lysis agent (Promega, Madison, WI, USA) after being incubated and treated with EGCG, PD and SP. Then, a luminometer (Centro XS lb960, Berthold Technologies, Oak Ridge, TN, USA) was used to measure the luciferase activity in the cell lysis supernatant.

### 4.6. Matrigel Cell Invasion Assay

The experiment for cell invasion was performed in a 10-well chemotaxis chamber (Neuro Probe, Gaithersburg, MD, USA) device with membranes with an 8-µM pore (Neuro Probe) in DMEM with 10% FBS as the chemoattractant in the lower chamber. In 200 μL of medium, T24 cells were planted in the upper chamber containing IL-1β, EGCG, non-specific IgG, and anti-uPAR antibody. To act as a chemoattractant, DMEM with 10% FBS was supplemented in the lower chamber. Twenty-four hours after incubation, non-invading cells on the upper surface of the membrane were scraped off using a cotton swab, and invading cells on the lower surface of the membrane were stained with a Diff-Quick kit (Becton-Dickinson, Franklin Lakes, NJ, USA). Following two rinses with distilled water and drying in air, the number of invading cells was counted using a phase-contrast microscope.

### 4.7. Measurement of Intracellular Hydrogen Peroxide (H_2_O_2_)

The production of intracellular H_2_O_2_ was measured using 5- and 6-carboxy-2′,7′-dichlorodihydrofluorescein diacetate (DCFDA; Molecular Probes, Eugene, OR, USA) [72]. In summary, the cells were cultured for 24 h in serum starved DMEM supplemented with 1% FBS. The cells were then switched to serum-free DMEM without phenol red and exposed to IL-1β for 30 min. Prior to the IL-1β treatment, the cells were pretreated with 10 to 50 μM EGCG for 1 h to determine how they would affect the ROS production caused by IL-1β. Then, after treatment with DCFDA (5 μg/mL) for 15 min, a laser scanning confocal microscope (Carl Zeiss, Germany) was used to quickly monitor the fluorescence excited at 488 nm using an argon laser and the emission at a 515 nm longpass filter.

### 4.8. Statistical Analysis

Each value represents three separate experiments and is presented as the mean ± standard deviation (SD). The results were expressed by using Graph Pad Prism software (Version 8.0). For the multivariable analyses, we used ANOVA with Tukey’s multiple comparison test; *p* < 0.05 (^#^, *), *p* < 0.01 (^##^, **), *p* < 0.001 (^###^, ***), and *p* < 0.0001 (^####^, ****) were considered statistically significant.

## 5. Conclusions

In this study, we investigated the roles and signaling pathways of green tea extract EGCG, that belong to the family of flavonoids with antioxidant properties and are associated with bladder cancer cell invasion. Our results demonstrate that EGCG inhibits IL-1β-induced uPAR expression by inhibiting ROS production and AP-1/NF-κB activation, and by decreasing cell invasion in T24 bladder cancer cells. According to these results, EGCG has an anticarcinogenic effect and may serve as a new therapeutic strategy for bladder cancer.

## Figures and Tables

**Figure 1 ijms-23-14008-f001:**
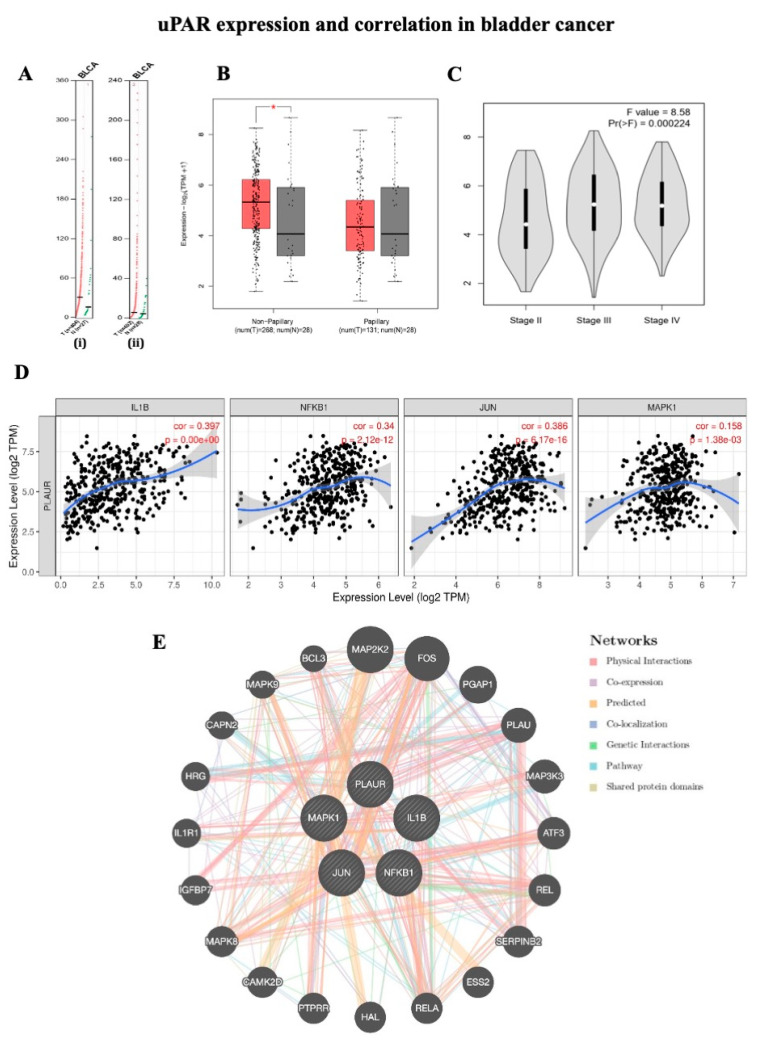
(**A**) Expression of uPAR (i) and IL1B (ii) in TPM scores of bladder cancer data obtained from GEPIA 2. (**B**) Expression of uPAR in non-papillary and papillary bladder cancer (*p* < 0.05). (**C**) uPAR expression in various stages of bladder cancer. Expression correlation and protein interaction between IL1B, NFKB1, JUN, and MAPK1 with PLAUR in BLCA data obtained from TIMER2.0 (*p* < 0.05) (**D**) and GeneMANIA (**E**).

**Figure 2 ijms-23-14008-f002:**
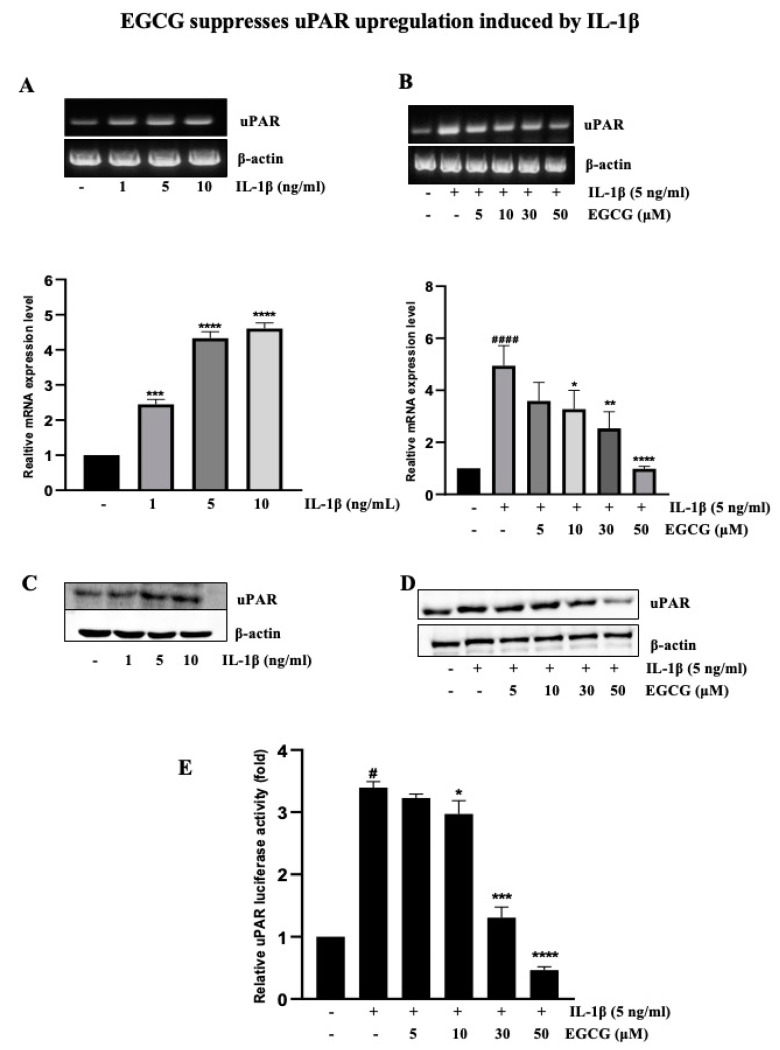
EGCG inhibits IL-1β-induced uPAR expression in T24 bladder cancer cells. (**A**) T24 cells were treated with 0−10 ng/mL of IL-1β for 4 h, and uPAR mRNA level was evaluated by RT-PCR. The bladder cancer cell line T24 was pretreated with different concentration of EGCG (5–50 μM) for 1 h. The cells were then incubated with IL-1β (5 ng/ ml) for 4 h. Then, mRNA was extracted and uPAR expression was evaluated by RT-PCR (**B**). (**C**,**D**) T24 cells were treated with 0–10 ng/mL IL-1β for 4 h, and uPAR protein level was evaluated by Western blotting. The T24 cell line was pretreated with different concentration of EGCG (5–50 μM) for 1 h. The cells were then incubated with IL-1β (5 ng/ ml) for 4 h. Then, protein was extracted and uPAR expression was evaluated by Western blotting. (**E**) T24 cells transfected with the pGL3-uPAR plasmid were pretreated with 5–50 μM for 1 h then incubated with 5 ng/ mL IL-1β for 12 h submitted for the luciferase assay. The above data represent the mean ± SD from triplicate measurements (^#^ *p* < 0.05, ^####^ *p* < 0.0001 versus control; * *p* < 0.05, ** *p* < 0.01, *** *p* < 0.001, **** *p* < 0.0001 versus IL-1β).

**Figure 3 ijms-23-14008-f003:**
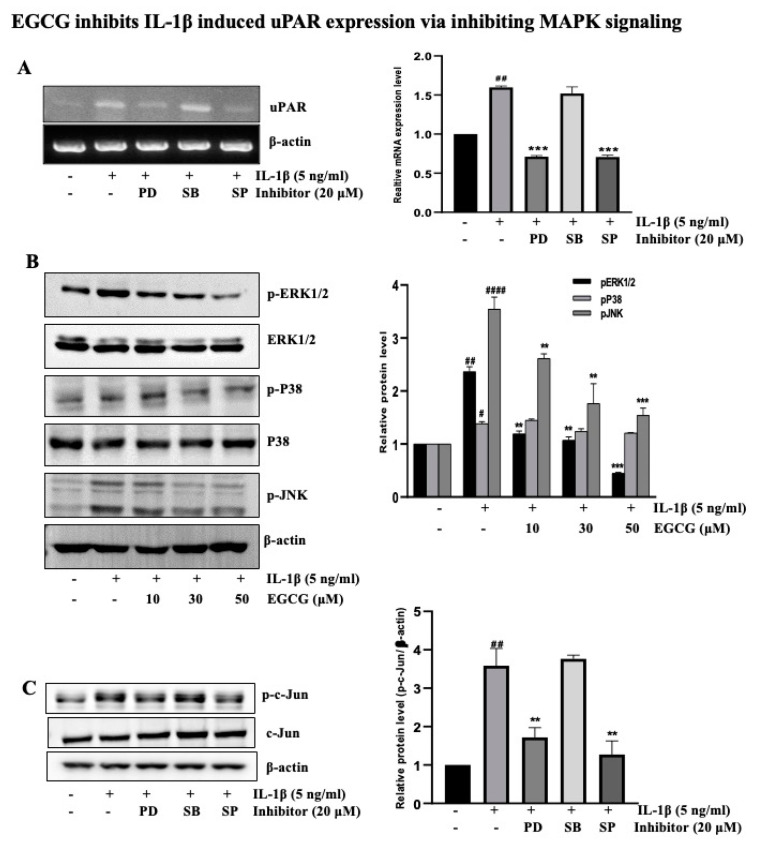
EGCG inhibits MAPK signaling stimulated by IL-1β in T24 bladder cancer cells (**A**) RT-PCR analysis to detect the uPAR expression by 1 h pretreatment with signaling inhibitor and then 5 ng/mL IL-1β treatment for 4 h. (**B**) T24 cells pretreated with 10–50 μM EGCG for 1 h were exposed to 5 ng/mL IL-1β for 30 min. The cells were then extracted for protein and tested for the level of phosphorylated ERK1/2, JNK and P38. (**C**) Phosphorylation of c-Jun by pre-treatment with MAPK inhibitors for 1 h were exposed to 5 ng/mL IL-1β for 30 min. The above data represent the mean ± SD from triplicate measurements (^#^ *p* < 0.05, ^##^ *p* < 0.01, ^####^ *p* < 0.0001 versus control; ** *p* < 0.01, *** *p* < 0.001 versus IL-1β).

**Figure 4 ijms-23-14008-f004:**
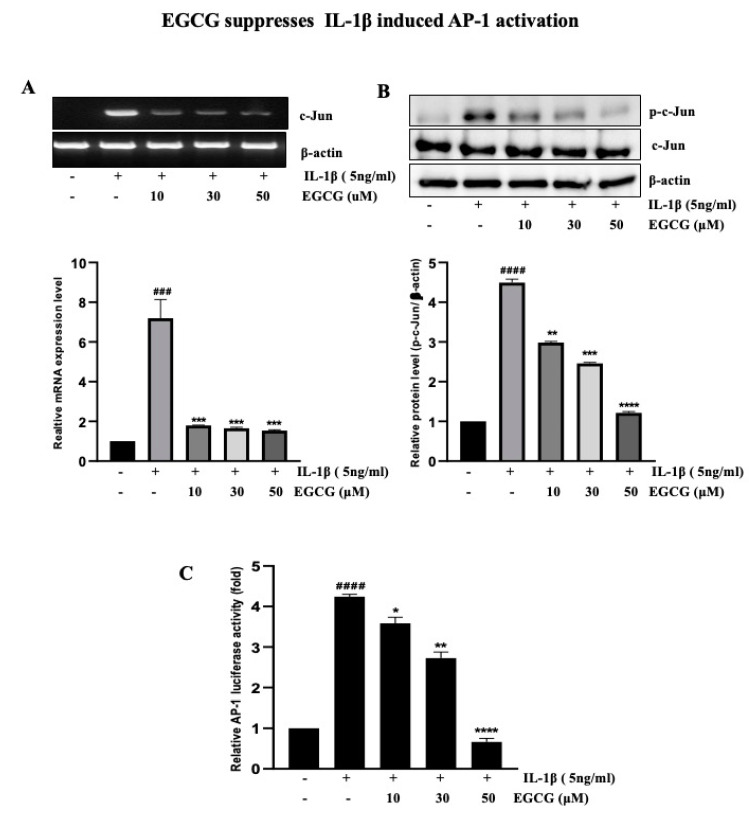
EGCG inhibits IL-1β-induced uPAR expression by suppressing the transcriptional activity of AP-1 in T24 cells. (**A**) T24 cells were pretreated with EGCG (10, 30, and 50 μM) for 1 h and then 5 ng/mL IL-1β for 4 h, and the cells were then extracted for mRNA to check uPAR expression by RT-PCR for the c-Jun expression. T24 cells were pretreated again with different concentrations of EGCG followed by 5 ng/mL IL-1β for 30 min of treatment and the phosphorylation of c-Jun expression by performing Western blotting (**B**). (**C**) Cells were pretreated with EGCG (10, 30, and 50 μM), were transiently transfected with the AP-1 luciferase reporter plasmid and were incubated with 5 ng/mL IL-1β; the cells were lysed, and luciferase activity was determined. The above data represent the mean ± SD from triplicate measurements (^###^ *p* < 0.001, ^####^ *p* < 0.0001 versus control; * *p* < 0.05, ** *p* < 0.01, *** *p* < 0.001, **** *p* < 0.0001 versus IL-1β).

**Figure 5 ijms-23-14008-f005:**
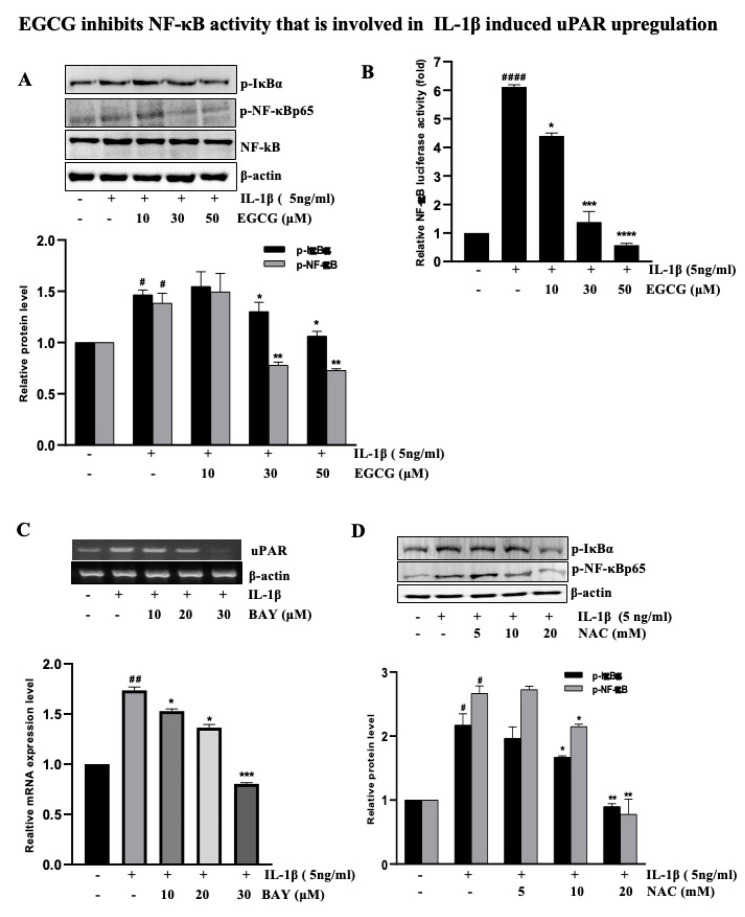
EGCG Suppresses IL-1β-induced NF-κB signaling in T24 cells. (**A**) T24 cells were pre-treated with 10–50 µM EGCG and incubated with 5 ng/mL IL-1β for 4 h; cells were then harvested to check for p65 and IκBα by Western blot analysis. (**B**) Cells were pretreated with EGCG (10, 30, and 50 μM), were transiently transfected with the AP-1 luciferase reporter plasmid and were incubated with 5 ng/mL IL-1β; the cells were lysed, and luciferase activity was determined. (**C**) T24 cells were pretreated with 10–20 µM Bay and incubated with 5 ng/mL IL-1β for 4 h. The cells were then extracted for mRNA, and uPAR expression was checked by RT-PCR. (**D**) T24 cells were pretreated with 5–20 mM NAC for 1 h and incubated with 5 ng/mL IL-1β for 4 h, cells were then harvested to check for p-NF-κB and p-IκBα by Western blot analysis. The above data represent the mean ± SD from triplicate measurements (^#^ *p* < 0.05, ^##^ *p* < 0.01, ^####^ *p* < 0.0001 versus control; * *p* < 0.05, ** *p* < 0.01, *** *p* < 0.001, **** *p* < 0.0001 versus IL-1β).

**Figure 6 ijms-23-14008-f006:**
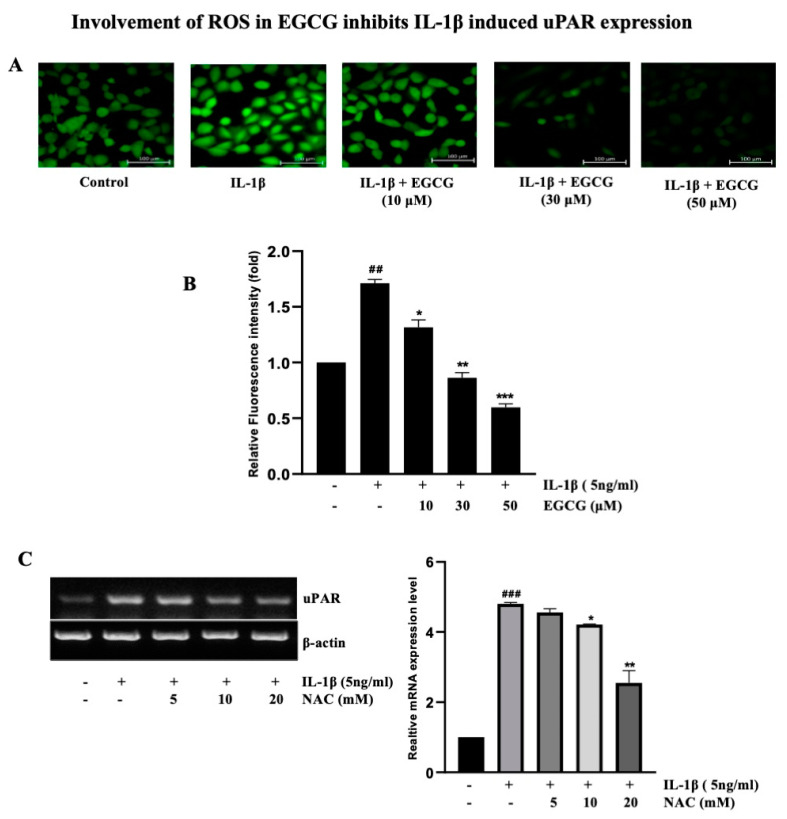
EGCG blocks ROS production induced by IL-1β in T24 cells. (**A**) Representative images (200X) and statistically quantitative values of ROS production by confocal microscope. (**B**) Relative fluorescence intensity fold by EGCG and IL-1β. (**C**) RT-PCR analysis for uPAR expression on NAC (5–20 mM) pretreatment for 1 h followed by 5 ng/mL IL-1β treatment for 4 h. The above data represent the mean ± SD from triplicate measurements (^##^ *p* < 0.01, ^###^ *p* < 0.001, versus control; * *p* < 0.05, ** *p* < 0.01, *** *p* < 0.001 versus IL-1β).

**Figure 7 ijms-23-14008-f007:**
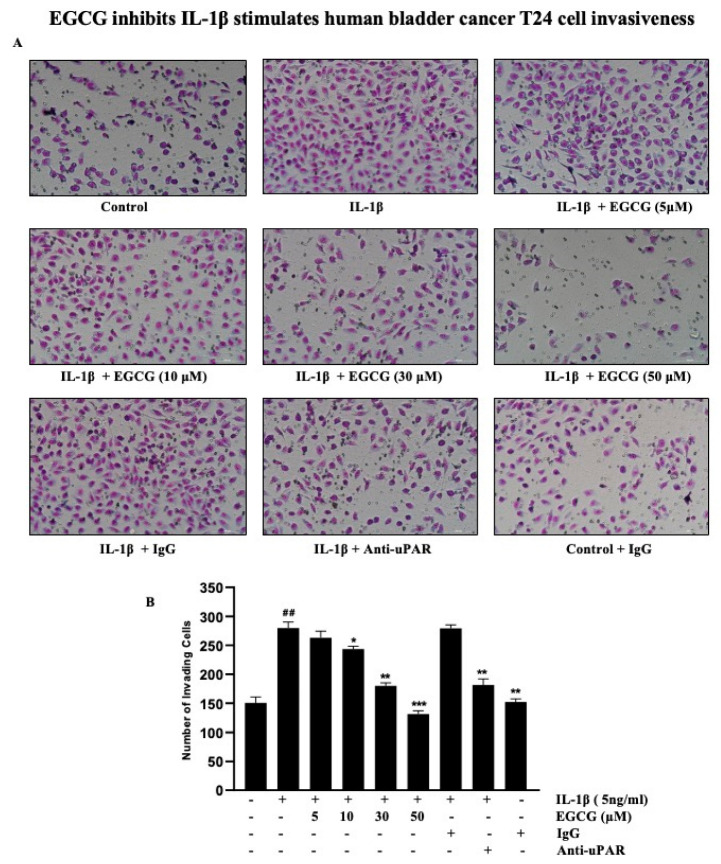
EGCG inhibits the invasion of T24 cells by suppressing uPAR expression. (**A**) T24 cells were incubated with 5 ng/mL IL-1β in the presence or absence of EGCG or 200 ng/mL, anti-uPAR antibody in a Matrigel apparatus for 24 h. (**B**) The cells were incubated with 5 ng/mL IL-1β in the presence of non-specific IgG (200 ng/mL), anti-uPAR antibody (200 ng/mL), and 5–50 μM EGCG. After incubation for 24 h, cells invading the undersurface of the chamber membrane were counted using a phase-contrast light microscope (20×) by staining with Diff-Quick stain. (^##^ *p* < 0.01 versus control; * *p* < 0.05, ** *p* < 0.01, *** *p* < 0.001 versus IL-1β).

**Figure 8 ijms-23-14008-f008:**
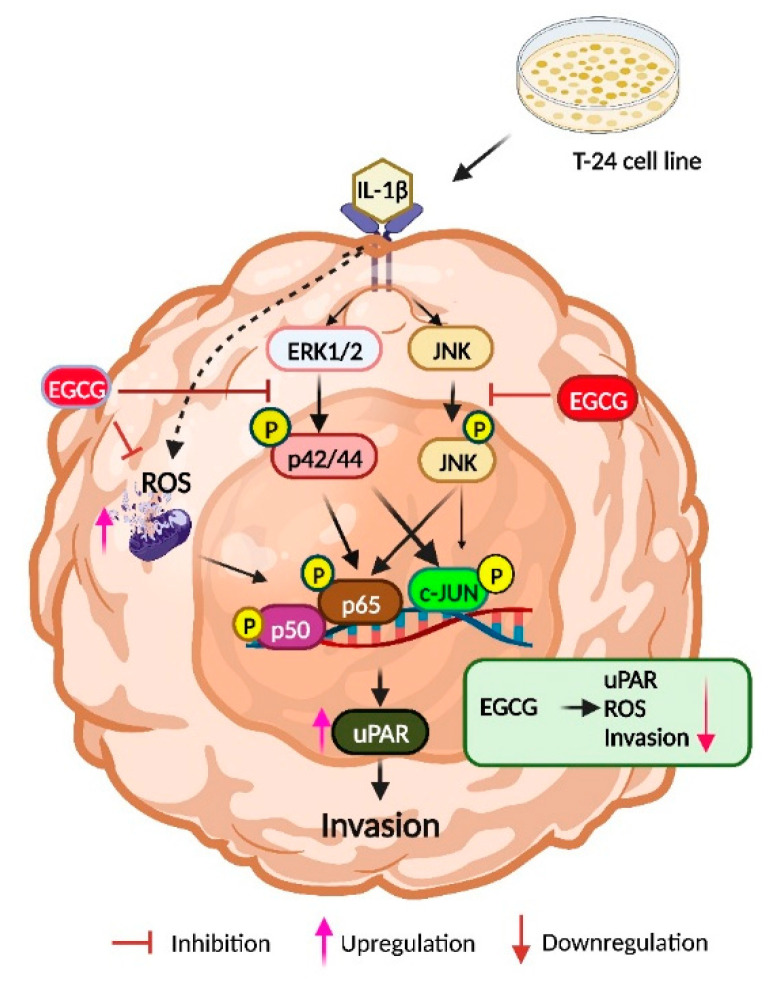
An overview of the process by which EGCG inhibits IL-1β’s induction of uPAR expression in T24 cells. By increasing uPAR expression, IL-1β stimulates the invasion of T24 cells. Through the ERK1/2/JNK signaling pathways in T24 cells, IL-1β stimulates the transcriptional activity of AP-1 and NF-κB, which in turn promotes uPAR expression. By preventing the production of uPAR and attenuating the (ERK1/2, JNK)/AP-1 and (ERK1/2, JNK)/ NF-κB signaling pathways, EGCG prevents the invasion of T24 cells that are triggered by IL-1β.

## Data Availability

Data are contained within the article.

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
