# Peer review of "(-)-Epigallocatechin-3-Gallate Prevents IL-1β-Induced uPAR Expression and Invasiveness via the Suppression of NF-κB and AP-1 in Human Bladder Cancer Cells"

_ijms, 2022, doi:10.3390/ijms232214008_

Round 1

Reviewer 1 Report

Sah et al. have studied the role of EGCG in preventing IL-1 b-induced uPAR-mediated invasiveness in bladder cancer. The authors use public databases for the rationale of targeting uPAR in cancer, and through in vitro experimental analysis, they confirm IL-1b mediates the upregulation of uPAR. This upregulation can be curtailed by treating the cells with EGCG. Further, the authors designed experiments using various inhibitors and showed that NF-kB and AP-1 could function as critical factors for IL-1 b-mediated uPAR upregulation. Finally, the authors show that EGCG does its anti-cancerous activity by scavenging ROS.

The reviewer has a few suggestions for improving the manuscript.

1. In line 232, the authors mention that "the results of an RT-PCR experiment revealed that ERK1/2 and JNK activation largely activates AP-1, a crucial  transcription factor for controlling uPAR expression" by figure 3A does not support the claim. The authors are suggested to perform more confirmatory experiments to show the direct role of AP1 in uPAR regulation under various treatment of inhibitors.

2. In figure 3C, the authors are requested to show the protein expression of uPAR under various treatment conditions. Rectify the molecular units in figure 3C.

3. In line 290, the authors mention that "NAC, which is an antioxidant drug that increases glutathione levels and scavenges free radicals, suppresses NF-kB activation induced by several extracellular stimuli, including oxidative stress." How is NAC relevant to this study when the study is on EGCG?

4. In line 294, "These results indicated that NAC inhibits NF- kB downstream signaling pathways," I think the authors mean that EGCG inhibits NF-kB downstream signaling in a similar manner to NAC. The authors are requested to edit the paragraph and focus on what the research question asked.

Minor comment

1. The authors are requested to edit all typos and follow a uniform style throughout the manuscript.

2. The authors are requested to provide better quality graph images as the legends of the axis are not readable in the current submitted version

Author Response

Reviewer 1: Thank you for the valuable suggestions to improving the manuscript

  1. In line 232, the authors mention that "the results of an RT- PCR experiment revealed that ERK1/2 and JNK activation largely activates AP-1, a crucial transcription factor for controlling uPAR expression" by figure 3A does not support the claim. The authors are suggested to perform more confirmatory experiments to show the direct role of AP1 in uPAR regulation under various treatment of inhibitors.

Response: A luciferase assay was done to confirm AP-1's regulation of uPAR, which is shown in Figure 4C and also shown in the supplementary data (Supplementary Figure 2- Line 241).

  1. In figure 3C, the authors are requested to show the protein expression of uPAR under various treatment conditions. Rectify the molecular units in figure 3C.

Response: The molecular units in figure 3C have been corrected

  1. In line 290, the authors mention that "NAC, which is an antioxidant drug that increases glutathione levels and scavenges free radicals, suppresses NF-kB activation induced by several extracellular stimuli, including oxidative stress." How is NAC relevant to this study when the study is on EGCG?

Response: NAC (N-acetyl-l-cysteine) is commonly used to identify and test ROS (reactive oxygen species) inducers, and to inhibit ROS. This study also focuses the EGCG inhibit the ROS production.

  1. In line 294, "These results indicated that NAC inhibits NF- kB downstream signaling pathways," I think the authors mean that EGCG inhibits NF-kB downstream signaling in a similar manner to NAC. The authors are requested to edit the paragraph and focus on what the research question asked.

 Response: As per the suggestions we edited the paragraph.

Minor comment

  1. The authors are requested to edit all typos and follow a uniform style throughout the manuscript.

 Response: We corrected the all-typo error in the manuscript

  1. The authors are requested to provide better quality graph images as the legends of the axis are not readable in the current submitted version

 Response: As per the reviewer suggestion, we given the better quality of graph images in revised version.

Reviewer 2 Report

The primary objective of this study was to examine the inhibitory effect of EGCG on IL-1β induced cell invasion by suppressing uPAR activity in T24 human bladder cancer cells. It was demonstrated that treatment with EGCG at 10 ~ 50 micro-M significantly suppressed IL-1β induced uPAR, p-ERK, p-JNK and p-c-Jun expression, NF-κB and AP-1 activation, ROS production and cell invasion in cultured T24 cells. It was concluded that the antitumor effect of EGCG was associated with its inhibitory effect on uPAR through the suppression of ERK, JNK, AP-1 and NF-κB activities.

Major points

1. Why was only one human bladder cancer cell line used in this study, and why was the T24 human bladder cancer cell line was used?

2. Why were EGCG concentrations used in the in vitro treatment at 10 ~ 50 micro-M? The authors should conduct the cytotoxicity assay (either MTT or SRB assay) to determine the IC50 of EGCG in T24 cells first and then select the concentrations for the in vitro treatment. Is the concentration of 10 micro-M of EGCG achievable in tumors in vivo?

3. What was the rationale of testing PD, SB, SP, BAY and NAC (Figure 3 and Figure 5)? If the authors wanted to compare the effect of EGCG with those compounds, it should be discussed in the Discussion section.

Minor points:

Line 213. EGCG concentrations should be 0~ 50 micro-M.

Line 177. Was any post-hoc multiple comparison test used following the ANOVA?

Author Response

Reviewer 2: Thank you for the valuable suggestions to improving the manuscript.

Major points

  1. Why was only one human bladder cancer cell line used in this study, and why was the T24 human bladder cancer cell line was used?

Response:  Transitional cell carcinoma (TCC) is the most common type of bladder cancer. T24 cells, a cell line established from a human urinary bladder cancer patient, are high-grade and invasive TCC. As part of our research, we investigated different therapeutic agents for bladder cancer. This study is a continuation of our previous studies. Therefore, we focused on a T24 cell line in order to provide concise and accurate results for bladder cancer.

  1. Why were EGCG concentrations used in the in vitro treatment at 10 ~ 50 micro-M? The authors should conduct the cytotoxicity assay (either MTT or SRB assay) to determine the IC50 of EGCG in T24 cells first and then select the concentrations for the in vitro treatment. Is the concentration of 10 micro-M of EGCG achievable in tumors in vivo?

Response: The MTT assay was performed, and the results are shown in the supplementary data (Supplementary Figure 1). In in-vivo study, Lambert et al., reported that pharmacokinetic absorption of EGCG. The study concluded that oral administration of 163.8 μmol/kg of EGCG with rats or 4.4 μmol/kg of EGCG with humans provided peak EGCG bioavailability in mice and humans, enabling the investigation of cancer prevention potential.

Reference: Lambert JD, Lee MJ, Lu H, Meng X, Hong JJ, Seril DN, Sturgill MG, Yang CS. Epigallocatechin-3-gallate is absorbed but extensively glucuronidated following oral administration to mice. The Journal of nutrition. 2003 Dec 1;133(12):4172-7.

  1. What was the rationale of testing PD, SB, SP, BAY and NAC (Figure 3 and Figure 5)? If the authors wanted to compare the effect of EGCG with those compounds, it should be discussed in the Discussion section.

Response: The rationale of testing PD, SB, SP, BAY and NAC was mentioned in line 230 – 234. As per reviewer suggestion, we included the discussion with various PD, SB, SP, BAY and NAC inhibitors.

Minor points:

Line 213. EGCG concentrations should be 0~ 50 micro-M.

Response: We corrected the EGCG concentrations units in line 213.

Line 177. Was any post-hoc multiple comparison test used to follow the ANOVA?

Response: Yes, we done the Tukey’s multiple comparison test used following the ANOVA and mentioned in line 180.

t

Round 2

Reviewer 2 Report

The authors have addressed all of my concerns.